# Deep Equilibrium Models

**Shaojie Bai**
Carnegie Mellon University

**J. Zico Kolter**
Carnegie Mellon University
Bosch Center for AI

**Vladlen Koltun**
Intel Labs

## Abstract

We present a new approach to modeling sequential data: the deep equilibrium model (DEQ). Motivated by an observation that the hidden layers of many existing deep sequence models converge towards some fixed point, we propose the DEQ approach that *directly* finds these equilibrium points via root-finding. Such a method is equivalent to running an *infinite* depth (weight-tied) feedforward network, but has the notable advantage that we can analytically backpropagate through the equilibrium point using implicit differentiation. Using this approach, training and prediction in these networks require only *constant* memory, regardless of the effective "depth" of the network. We demonstrate how DEQs can be applied to two state-of-the-art deep sequence models: self-attention transformers and trellis networks. On large-scale language modeling tasks, such as the WikiText-103 benchmark, we show that DEQs 1) often improve performance over these state-of-the-art models (for similar parameter counts); 2) have similar computational requirements to existing models; and 3) vastly reduce memory consumption (often the bottleneck for training large sequence models), demonstrating an up-to *88%* memory reduction in our experiments. The code is available at `https://github.com/locuslab/deq`.

## 1   Introduction

Most modern feedforward deep networks are built on the core concept of *layers*. In the forward pass, each network consists of a stack of some $L$ transformations, where $L$ is the depth of the network. To update these networks, the backward passes rely on backpropagating through the same $L$ layers via the chain rule, which typically necessitates that we store the intermediate values of these layers. The value for $L$ is usually a hyperparameter and is picked by model designers (e.g., ResNet-101 [25]). Among the many applications of deep networks, sequence modeling has witnessed continuous advances in model architectures. Specifically, while recurrent networks have long been the dominant model for sequences [21, 26, 14, 34], deep feedforward architectures based on temporal convolutions [49, 47, 7] and self-attention [48, 16, 13] have (re-)emerged to claim superior performance on a variety of sequence prediction tasks.

In very general terms, a deep feedforward sequence model can be written as the following iteration:

$$\mathbf{z}_{1:T}^{[i+1]} = f_\theta^{[i]}\big(\mathbf{z}_{1:T}^{[i]}; \mathbf{x}_{1:T}\big) \quad \text{for } i = 0, 1, 2, \ldots, L-1 \tag{1}$$

where $i$ is the layer index; $\mathbf{z}_{1:T}^{[i]}$ is the hidden sequence of length $T$ at layer $i$; $\mathbf{x}_{1:T}$ is the input sequence (i.e., we are choosing to explicitly model skip connections, for reasons we explain later); and $f_\theta^{[i]}$ is some nonlinear transformation which typically enforces causality (i.e., future time points cannot influence past ones). Our paper derives its motivation from surprising recent works that employ the *same* transformation in each layer (known as *weight tying*, with $f_\theta^{[i]} = f_\theta, \forall i$) and still achieve results competitive with the state-of-the-art [18, 8, 15]. This raises an interesting question: If the same transformation is applied at each layer of a deep network, what is the limit of this process, and how do we model it?

In this paper, we propose a new approach to "deep" modeling that addresses this question. Specifically, we introduce the deep equilibirum model (DEQ), a method that directly computes the fixed point $\mathbf{z}_{1:T}^{\star}$ of a nonlinear transformation, i.e., the solution to the nonlinear system

$$\mathbf{z}_{1:T}^{\star} = f_\theta(\mathbf{z}_{1:T}^{\star}; \mathbf{x}_{1:T}). \tag{2}$$

This solution corresponds to the eventual hidden layer values of an *infinite depth* network. But instead of finding this value by iterating the model, we propose to directly (and in practice, more quickly) solve for the equilibrium via any black-box root-finding method. Importantly, we show that DEQ can *directly* differentiate through the fixed point equations via implicit differentation, which does not require storing *any* intermediate activation values. In other words, we can backpropagate through the infinite-depth network while using only *constant* memory, equivalent to a single layer's activations.

After developing the generic DEQ approach, we study in detail the instantiation of DEQ via two feedforward sequence models: *trellis networks* (weight-tied temporal convolutions) [8] and memory-augmented *universal transformers* (weight-tied multi-head self-attention) [18, 16], both of which have obtained state-of-the-art performance (SOTA) on various sequence tasks. We show how both the forward and backward passes can be implemented efficiently via quasi-Newton methods. Finally, we demonstrate via experiments on large-scale high-dimensional sequence modeling benchmarks (e.g., WikiText-103 language modeling) that, despite only using constant memory, DEQ can attain modeling accuracy on par with (or even slightly better than) corresponding layer-based networks. We believe that DEQ offers a novel perspective on the analysis of sequential data.

## 2 Background

**Deep sequence models.** Given an input sequence $\mathbf{x}_{1:T} = [x_1, \ldots, x_T] \in \mathbb{R}^{T \times p}$, where $x_i \in \mathbb{R}^p$ (e.g., a word embedding) and $T \in \mathbb{N}$ is the sequence length, we define a sequence model as any function $G$ that produces output $G(\mathbf{x}_{1:T}) = \mathbf{y}_{1:T} = \in \mathbb{R}^{T \times q}$ that satisfies the causality constraint: $y_t$ depends only on $\mathbf{x}_{1:t}$ and not on any element of $\mathbf{x}_{t+1:T}$. Recent progress on autoregressive sequence tasks has been based on deep learning, where three major families of sequence models stand out. Recurrent networks (RNNs) [21, 51] as well as their variants such as LSTM [26] are universally applied and optimized in a variety of time-series tasks [9, 22, 34]. Alternatively, prior work has shown that deeply stacked temporal convolutions [49, 47, 17, 7] can achieve competitive results, especially on long sequences. Finally, the self-attention transformer architecture [48, 16] has also achieved SOTA on several NLP benchmarks [19, 13]. Efforts have also been devoted to drawing deeper connections among the three model families. Bai et al. [8] study the underlying relationship between RNNs and ConvNets, unifying these in the Trellis Network, which combines the benefits of both families. Dehghani et al. [18] introduce a recurrently-stacked universal transformer and demonstrate its effectiveness on text understanding and generation.

**Memory-efficient deep networks.** An important factor that limits the development of high-capacity networks is limited memory on hardware devices used for training. To address this issue, [12] proposes gradient checkpointing that reduces an $L$-layer network's memory requirement to $O(\sqrt{L})$ at the cost of extra forward passes (i.e., extra computations). Alternatively, [23, 30] develop reversible networks, where each layer's activations can be reconstructed from the next layer during backpropagation to reduce memory requirements. DEQs reduce memory consumption to a *constant* (i.e., independent of network "depth") by directly differentiating through the equilibrium point and thus circumventing the construction and maintenance of "layers".

**Continuous view of deep networks.** Some prior works have studied continuous views of deep networks. [41] proposes a biologically inspired equilibrium propagation framework for an energy-based model whose prediction is the fixed-point of the energy dynamics at its local minimum. [24, 11] model deep ResNets by black-box ODE solvers in forward and backward passes (as if the network has smaller "layer steps") given the start- and end-points of a dynamical system. For deep sequence models, [43, 36] consider the RNN as a dynamical system to investigate its stability properties.

Our work takes a further step in the direction of the aforementioned areas. While some of the prior work has primarily focused on the analysis of residual architectures or small symmetric-weight energy-based models, our work is not predicated on any specific type of interlayer transformation. We show that DEQs can be easily instantiated via two very different sequence learning architectures. More fundamentally, unlike ODE-based methods, which use the adjoint system to backpropagate

through the entire latent trajectory, the DEQ model solves directly for sequence-level equilibria via a quasi-Newton method and backpropagates directly through this fixed point, without regard for the solution path that brought it there. Moreover, while ODE-based models [24, 11] were verified on numerical experiments and MNIST classification, computation and numerical stability issues challenge their application to large-scale problems. In comparison, we demonstrate the applicability of DEQs on realistic high-dimensional sequence tasks with competitive performance, while enjoying similar constant-memory benefits as [11].

**Implicit layers in deep learning.** The DEQ model can be viewed as an infinitely deep network, but interestingly can also be viewed as a *single*-layer network, with the caveat that the layer is defined *implicitly*: the output $\mathbf{z}_{1:T}^\star$ is defined as the value which solves some non-linear equation. There has been a growing interest in implicit layers in recent years [37, 3, 37, 50], but the precise formulation of the DEQ is quite different, and our current models represent the largest-scale practical application of implicit layers in deep learning of which we are aware. Concurrent work [20] also looks at such implicit layers in a broad sense and focuses on training small models via Lagrangian methods; a combination of these approaches with the DEQ model is a promising avenue for future work.

Another thread of work on implicit layers traces back to some of the original papers on recurrent networks trained via recurrent backpropagation (RBP) [2, 38]. Recent work [28] has re-examined RBP and established an implicit, constant-memory variant based on conjugate gradient and Neumann series. A number of related papers also enforce fixed point conditions within RNN architectures [54, 27]. Whereas the DEQ model shares similarities with the RBP approach, some major differences involve: 1) the explicit use of equilibrium as a replacement for depth in general networks, along with our proof of the universality of these models to replace depth; 2) the use of the approach in methods outside of fixed-input RNNs (i.e., same input vector $x_t$ for all $t$), especially the compatibility with SOTA architectures; and 3) the scalability of the DEQ model to practical tasks where it achieves results on par with the current SOTA, whereas RBP has typically been applied in small-scale settings.

## 3 The Deep Equilibrium Sequence Model

We broadly consider the class of *weight-tied* deep sequence models (with passthrough connections from the input to each layer), which consist of the update

$$\mathbf{z}_{1:T}^{[i+1]} = f_\theta(\mathbf{z}_{1:T}^{[i]}; \mathbf{x}_{1:T}), \quad i = 0, \dots, L-1, \quad \mathbf{z}_{1:T}^{[0]} = \mathbf{0}, \quad G(\mathbf{x}_{1:T}) \equiv \mathbf{z}_{1:T}^{[L]} \tag{3}$$

We note that this model encapsulates classes such as the trellis network [8] and the universal transformer [18] (which is typically not written with passthrough connections, but this is a trivial modification). Such weight-tying is generally considered to come with four major benefits: 1) it acts as a form of regularization that stabilizes training and supports generalization; 2) it significantly reduces the model size; 3) it is trivial to show that *any* deep network can be represented by a weight-tied deep network of equal depth and only a linear increase in width (see Appendix C); and 4) the network can be unrolled to *any* depth, typically with improved feature abstractions as depth increases [8, 18]. However, in practice almost all such models (and deep nets in general) are stacked, trained and evaluated *by unrolling a pre-determined, fixed number of layers*. One reason is the limited memory on training hardware: the models need to store intermediate hidden units for backpropagation and thus cannot be trained beyond a certain depth that depends on the available memory.

In principle, the network could have infinite depth. This is attained in the limit of unrolling a weight-tied model for an ever higher number of layers. What is the limit of this process? In practice, for certain classes of $f_\theta$ (discussed later), we hypothesize and observe that such weight-tied models tend to converge to a *fixed point* as depth increases towards infinity (see Appendix D for empirical evidence). In other words, as each layer refines the previous one by combining temporal features across the sequence, increasing depth towards infinity brings "diminishing returns": each additional layer has a smaller and smaller contribution until the network reaches an equilibrium:

$$\lim_{i \to \infty} \mathbf{z}_{1:T}^{[i]} = \lim_{i \to \infty} f_\theta\big(\mathbf{z}_{1:T}^{[i]}; \mathbf{x}_{1:T}\big) \equiv f_\theta\big(\mathbf{z}_{1:T}^\star; \mathbf{x}_{1:T}\big) = \mathbf{z}_{1:T}^\star \tag{4}$$

### 3.1 The Deep Equilibrium Approach

We introduce the deep equilibrium model (DEQ) which, instead of iteratively stacking $f_\theta$, directly solves for and differentiates through the equilibrium state.

### 3.1.1 Forward Pass

Unlike a conventional network where the output is the activations from the $L^{\text{th}}$ layer, the output of a DEQ is the equilibrium point itself. Therefore, the forward evaluation could be any procedure that solves for this equilibrium point. Conventional deep sequence networks, if they converge to an equilibrium, can be considered a form of *fixed-point iterations*:

$$\mathbf{z}_{1:T}^{[i+1]} = f_\theta\big(\mathbf{z}_{1:T}^{[i]}; \mathbf{x}_{1:T}\big) \quad \text{for } i = 0, 1, 2, \ldots \tag{5}$$

One can alternatively use other methods that provide faster convergence guarantees. For notational convenience, we define $g_\theta$ and rewrite Eq. (4) as $g_\theta(\mathbf{z}_{1:T}^\star; \mathbf{x}_{1:T}) = f_\theta\big(\mathbf{z}_{1:T}^\star; \mathbf{x}_{1:T}\big) - \mathbf{z}_{1:T}^\star \to 0$. The equilibrium state $\mathbf{z}_{1:T}^\star$ is thus the root of $g_\theta$, which we can find more easily with Newton's method or quasi-Newton methods (e.g., Broyden's method [10]):

$$\mathbf{z}_{1:T}^{[i+1]} = \mathbf{z}_{1:T}^{[i]} - \alpha B g_\theta(\mathbf{z}_{1:T}^{[i]}; \mathbf{x}_{1:T}) \quad \text{for } i = 0, 1, 2, \ldots \tag{6}$$

where $B$ is the Jacobian inverse (or its low-rank approximation) at $\mathbf{z}_{1:T}^{[i]}$, and $\alpha$ is the step size. But generally, one can exploit any black-box root-finding algorithm to solve for the equilibrium point in the forward pass, given an initial estimate $\mathbf{z}_{1:T}^{[0]}$ (which we set to $\mathbf{0}$): $\mathbf{z}_{1:T}^\star = \mathsf{RootFind}(g_\theta; \mathbf{x}_{1:T})$

### 3.1.2 Backward Pass

A major problem with using a black-box RootFind is that we are no longer able to rely on explicit backpropagation through the exact operations in the forward pass. While one can certainly fix an algorithm (say Newton's method) to obtain the equilibrium, and then store and backpropagate through all the Newton iterations, we provide below an alternative procedure that is much simpler, requires constant memory, and assumes no knowledge of the black-box RootFind.

**Theorem 1.** *(**Gradient of the Equilibrium Model**) Let $\mathbf{z}_{1:T}^\star \in \mathbb{R}^{T \times d}$ be an equilibrium hidden sequence with length $T$ and dimensionality $d$, and $\mathbf{y}_{1:T} \in \mathbb{R}^{T \times q}$ the ground-truth (target) sequence. Let $h : \mathbb{R}^d \to \mathbb{R}^q$ be any differentiable function and let $\mathcal{L} : \mathbb{R}^q \times \mathbb{R}^q \to \mathbb{R}$ be a loss function (where $h, \mathcal{L}$ are applied in a vectorized manner) that computes*

$$\ell = \mathcal{L}(h(\mathbf{z}_{1:T}^\star), \mathbf{y}_{1:T}) = \mathcal{L}(h(\mathsf{RootFind}(g_\theta; \mathbf{x}_{1:T})), \mathbf{y}_{1:T}). \tag{7}$$

*Then the loss gradient w.r.t. $(\cdot)$ (for instance, $\theta$ or $\mathbf{x}_{1:T}$) is*

$$\frac{\partial \ell}{\partial (\cdot)} = -\frac{\partial \ell}{\partial \mathbf{z}_{1:T}^\star}\big(J_{g_\theta}^{-1}\big|_{\mathbf{z}_{1:T}^\star}\big)\frac{\partial f_\theta(\mathbf{z}_{1:T}^\star; \mathbf{x}_{1:T})}{\partial (\cdot)} = -\frac{\partial \ell}{\partial h}\frac{\partial h}{\partial \mathbf{z}_{1:T}^\star}\big(J_{g_\theta}^{-1}\big|_{\mathbf{z}_{1:T}^\star}\big)\frac{\partial f_\theta(\mathbf{z}_{1:T}^\star; \mathbf{x}_{1:T})}{\partial (\cdot)}, \tag{8}$$

*where $J_{g_\theta}^{-1}\big|_{\mathbf{x}}$ is the inverse Jacobian of $g_\theta$ evaluated at $\mathbf{x}$.*

The proof is provided in Appendix A. The insight provided by Theorem 1 is at the core of our method and its various benefits. Importantly, the backward gradient through the "infinite" stacking can be represented as one step of matrix multiplication that involves the Jacobian at equilibrium. For instance, an SGD update step on model parameters $\theta$ would be

$$\theta^+ = \theta - \alpha \cdot \frac{\partial \ell}{\partial \theta} = \theta + \alpha \frac{\partial \ell}{\partial \mathbf{z}_{1:T}^\star}\big(J_{g_\theta}^{-1}\big|_{\mathbf{z}_{1:T}^\star}\big)\frac{\partial f_\theta(\mathbf{z}_{1:T}^\star; \mathbf{x}_{1:T})}{\partial \theta}. \tag{9}$$

Note that this result is independent of the root-finding algorithm we choose or the internal structure of the transformation $f_\theta$, and thus does not require any storage of the intermediate hidden states, which is necessary for backpropagation in conventional deep networks.

### 3.1.3 Accelerating DEQ by Approximating the Inverse Jacobian

One challenge of enforcing the forward and backward passes described in Sections 3.1.1 and 3.1.2 is the cost of computing the exact inverse Jacobian $J_{g_\theta}^{-1}$ at every intermediate Newton iteration. We propose to address this using Broyden's method [10], a quasi-Newton approach that makes low-rank updates to approximate $J_{g_\theta}^{-1}$ via the Sherman-Morrison formula [42]:

$$J_{g_\theta}^{-1}\big|_{\mathbf{z}_{1:T}^{[i+1]}} \approx B_{g_\theta}^{[i+1]} = B_{g_\theta}^{[i]} + \frac{\Delta \mathbf{z}^{[i+1]} - B_{g_\theta}^{[i]}\Delta g_\theta^{[i+1]}}{\Delta \mathbf{z}^{[i+1]\top} B_{g_\theta}^{[i]}\Delta g_\theta^{[i+1]}}\Delta \mathbf{z}^{[i+1]\top} B_{g_\theta}^{[i]}, \tag{10}$$

where $\Delta\mathbf{z}^{[i+1]} = \mathbf{z}_{1:T}^{[i+1]} - \mathbf{z}_{1:T}^{[i]}$ and $\Delta g_\theta^{[i+1]} = g_\theta(\mathbf{z}_{1:T}^{[i+1]}; \mathbf{x}_{1:T}) - g_\theta(\mathbf{z}_{1:T}^{[i]}; \mathbf{x}_{1:T})$. Initially, we set $B_{g_\theta}^{[0]} = -I$ and the Broyden iterations are stopped when either the norm of $g_\theta^{[i]}$ falls below a tolerance $\varepsilon$ or when the maximum number of iterations is reached. This lets us avoid the cubic cost induced by the inverse operation.

A similar idea can be used for the backward pass as well. Specifically, to compute $-\frac{\partial\ell}{\partial\mathbf{z}_{1:T}^\star}\left(J_{g_\theta}^{-1}\big|_{\mathbf{z}_{1:T}^\star}\right)$ in Theorem 1, we can alternatively solve the linear system

$$\left(J_{g_\theta}^\top\big|_{\mathbf{z}_{1:T}^\star}\right)\mathbf{x}^\top + \left(\frac{\partial\ell}{\partial\mathbf{z}_{1:T}^\star}\right)^\top = \mathbf{0}, \tag{11}$$

where the first term (a vector-Jacobian product) can be efficiently computed via autograd packages (e.g., PyTorch [45]) for any $\mathbf{x}$, without explicitly writing out the Jacobian matrix. Such linear systems can generally be solved by any *indirect* methods that leverage fast matrix-vector products; we thus propose to also rely on Broyden's method (other indirect methods would also suffice) to solve for Eq. (11) and directly backpropagate through the equilibrium by Theorem 1 in the backward pass.

## 3.2 Properties of Deep Equilibrium Models

Section 3.1 develops a sequence model that, while still based on the deep learning philosophy, is quite different from other approaches in the field, as its output is agnostic to the choice of the RootFind algorithm in the forward pass. We now discuss some implications of the DEQ approach.

**Memory cost of DEQ.**   An important benefit of DEQ is its extreme memory efficiency. As outlined in Section 3.1.3, since we are able to use any root-finding algorithm for both the forward and backward passes (e.g., Broyden's method [10]), a DEQ only needs to store $\mathbf{z}_{1:T}^\star$ (the equilibrium sequence), $\mathbf{x}_{1:T}$ (input-related, layer-independent variables), and $f_\theta$ for the backward pass. Note that as we only need the vector-Jacobian product (with dimension $N \times Td$, where $N$ is the minibatch size) in Eq. (11), we never need to explicitly construct the Jacobian $J_{g_\theta}^\top\big|_{\mathbf{z}_{1:T}^\star}$, which could be prohibitively large on long and high-dimensional sequences (with dimension $N \times (Td)^2$). Compared to other deep networks, DEQs therefore offer a constant-memory alternative that enables models that previously required multiple GPUs and other implementation-based techniques (e.g., half-precision or gradient checkpointing [12, 13]) to fit easily into a single GPU.

**The choice of $f_\theta$.**   Our analysis in Sections 3.1.1, 3.1.2, and 3.1.3 is independent of the choice of $f_\theta$, and the same kind of memory benefit is present regardless of the type of $f_\theta$. However, to find the equilibrium in a reliable and efficient manner, generally $f_\theta$ needs to be stable and constrained. The two instantiations we provide in Section 4 are examples of stable transformations. (The gated activation in TrellisNet and layer normalization in the transformer constrain the output ranges.)

**Stacking the DEQ?**   A natural question arises: if one DEQ is good, can we get additional benefits by "stacking" DEQs (with potentially *different* classes of transformations)? The answer, somewhat surprisingly, is no, as evidenced by the following theorem, which is proved in Appendix B. The theorem essentially shows that stacking multiple DEQs does not create extra representational power over a single DEQ.

**Theorem 2.** *(Universality of "single-layer" DEQs.) Let $\mathbf{x}_{1:T} \in \mathbb{R}^{T\times p}$ be the input sequence, and $\theta^{[1]}, \theta^{[2]}$ the sets of parameters for stable transformations $f_{\theta^{[1]}} : \mathbb{R}^r \times \mathbb{R}^p \to \mathbb{R}^r$ and $v_{\theta^{[2]}} : \mathbb{R}^d \times \mathbb{R}^r \to \mathbb{R}^d$, respectively. Then there exists $\Gamma_\Theta : \mathbb{R}^{d+r} \times \mathbb{R}^p \to \mathbb{R}^{d+r}$, where $\Theta = \theta^{[1]} \cup \theta^{[2]}$, s.t.*

$$\mathbf{z}_{1:T}^\star = \mathsf{RootFind}\left(g_{\theta^{[2]}}^f; \mathsf{RootFind}\left(g_{\theta^{[1]}}^v; \mathbf{x}_{1:T}\right)\right) = \mathsf{RootFind}\left(g_\Theta^\Gamma; \mathbf{x}_{1:T}\right)_{[:, -d:]}, \tag{12}$$

*where $[\cdot]_{[:, -d:]}$ denotes the last $d$ feature dimensions of $[\cdot]$.*

## 4   Instantiations of DEQ

While the forward and backward analyses of DEQ do not depend on the internal structure of $f_\theta$, in this section we briefly highlight two examples of $f_\theta$ as specific instantiations of DEQ. Both models (TrellisNet [8] and self-attention [48, 18]) achieve state-of-the-art results on various sequence modeling benchmarks. Importantly, through these two very different models and their properties, we illustrate the compatibility of the DEQ approach with all three major families of existing deep sequence networks: *transformers*, *RNNs*, and *temporal convolutional networks (TCNs)*.

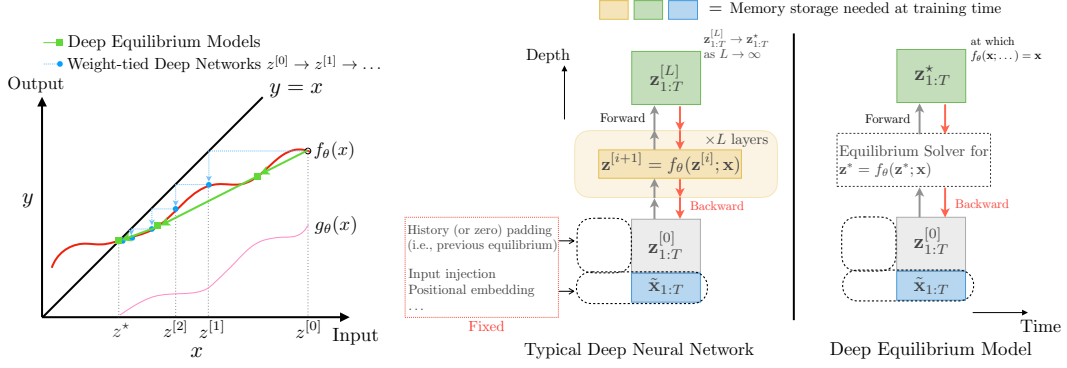

(a) A simple illustration of solving for an equilibrium point in 2D.

(b) A deep equilibrium model operates with significantly less memory than conventional deep nets due to an analytical backward pass.

Figure 1: Comparison of the DEQ with conventional weight-tied deep networks.

**Trellis networks.** We briefly introduce the trellis network (TrellisNet) here and refer interested readers to [8] for a detailed description. Generally, TrellisNet is a TCN with two modifications. First, a linear transformation of the original input sequence $\mathbf{x}_{1:T}$ is added to the convolutional outputs at all layers. Second, the convolutional kernel weights are tied across the depth of the network (i.e., TrellisNet is a weight-tied TCN). Thus we can write TrellisNet with convolutional kernel size $k$, dilation $s$, and nonlinearity $\psi$ in DEQ form as

$$\tilde{\mathbf{x}}_{1:T} = \text{Input injection (i.e., linearly transformed inputs by Conv1D}(\mathbf{x}_{1:T}; W_x))$$
$$f_\theta(\mathbf{z}_{1:T}; \mathbf{x}_{1:T}) = \psi(\text{Conv1D}([\mathbf{u}_{-(k-1)s:}, \mathbf{z}_{1:T}]; W_z) + \tilde{\mathbf{x}}_{1:T})$$

where $\mathbf{u}_{-(k-1)s:}$ is typically: 1) the last $(k-1)s$ elements of the previous sequence's output (if using history padding [8]); or 2) simply zero-padding. $[\cdot, \cdot]$ means concatenation along the temporal dimension. Following [8], we use the LSTM gated activation for $\psi$.

**Weight-tied transformers.** At a high level, multi-head self-attention transformers [48] are very different from most deep networks. Instead of convolutions or recurrence, a self-attention layer maps the input into $Q$ (query), $K$ (key), and $V$ (value) and computes the attention score between time-steps $t_i$ and $t_j$ as $[QK^\top]_{i,j}$. This attention score is then normalized via softmax and multiplied with the $V$ sequence to produce the output. Since the transformer is order-invariant, prior work proposed to add positional embeddings (PE) [48, 16] to the self-attention operation. Following this design, [18] further proposed the *universal transformer*, which "recurrently stacks" the transformer's self-attention and transition function block $\phi$ through a number of layers. Referring readers to [48, 16, 18] for more details, we write a weight-tied transformer in the DEQ form as

$$\tilde{\mathbf{x}}_{1:T} = \text{Input injection (i.e., linearly transformed inputs by } \mathbf{x}_{1:T} W_x)$$
$$f_\theta(\mathbf{z}_{1:T}; \mathbf{x}_{1:T}) = \text{LN}(\phi(\text{LN}(\text{SelfAttention}(\mathbf{z}_{1:T} W_{QKV} + \tilde{\mathbf{x}}_{1:T}; \text{PE}_{1:T}))))$$

where $W_{QKV} \in \mathbb{R}^{d \times 3d}$ produces the $Q, K, V$ for the multi-head self-attention, and LN stands for layer normalization [5]. Note that we add input injection $\tilde{\mathbf{x}}_{1:T}$ to $Q, K, V$ in addition to the positional embedding and initialize with $\mathbf{z}_{1:T}^{[0]} = \mathbf{0}$. Following prior work [48, 19, 16, 18], we use a 2-layer positionwise feedforward residual block for $\phi$. In our implementation, we use the memory-augmented transformer proposed by [16], where we feed $[\mathbf{z}_{-T':}^\star, \mathbf{z}_{1:T}]$ (i.e., with history padding of length $T'$) and relative positional embedding $\text{PE}_{-T':T}$ to the self-attention operation.

Figure 1 provides a generic comparison between these conventional weight-tied deep networks and the DEQ approach, highlighting the constant memory requirements of the latter.

## 5 Experiments

We evaluate DEQ on both synthetic stress tests and realistic large-scale language modeling (where complex long-term temporal dependencies are involved). We use the two aforementioned instantiations of $f_\theta$ in DEQ. On both WikiText-103 [35] (which contains $>$100M words and a vocabulary size of $>$260K) and the smaller Penn Treebank corpus (where stronger regularizations are needed for

Table 1: DEQ achieves strong performance on the long-range copy-memory task.

| | Models (Size) | | | |
| --- | --- | --- | --- | --- |
| | **DEQ-Transformer (ours)** (14K) | TCN [7] (16K) | LSTM [26] (14K) | GRU [14] (14K) |
| Copy Memory $T$=400 Loss | **3.5e-6** | **2.7e-5** | 0.0501 | 0.0491 |

Table 2: DEQ achieves competitive performance on word-level Penn Treebank language modeling (on par with SOTA results, without fine-tuning steps [34]). †The memory footprints are benchmarked (for fairness) on input sequence length 150 and batch size 15, which does not reflect the actual hyperparameters used; the values also do *not* include the memory for word embeddings.

| Word-level Language Modeling w/ Penn Treebank (PTB) | | | | |
| --- | --- | --- | --- | --- |
| Model | # Params | Non-embedding model size | Test perplexity | Memory† |
| Variational LSTM [22] | 66M | - | 73.4 | - |
| NAS Cell [55] | 54M | - | 62.4 | - |
| NAS (w/ black-box hyperparameter tuner) [32] | 24M | 20M | 59.7 | - |
| AWD-LSTM [34] | 24M | 20M | 58.8 | - |
| DARTS architecture search (second order) [29] | 23M | 20M | **55.7** | - |
| 60-layer TrellisNet (w/ auxiliary loss, w/o MoS) [8] | 24M | 20M | 57.0 | 8.5GB |
| **DEQ-TrellisNet (ours)** | 24M | 20M | 57.1 | **1.2GB** |

conventional deep nets) for word-level language modeling, we show that DEQ achieves competitive (or better) performance even when compared to SOTA methods (of the same model size, both weight-tied and not) while using significantly less memory. We provide a more detailed introduction of the tasks and datasets in Appendix F.

**Setting.** Both instantiations of DEQ use Broyden's method [10] to avoid direct computation of the inverse Jacobian, as described in Section 3.1.3. We note that the use of DEQ implicitly introduces a new "hyperparameter" – the stopping criterion for Broyden iterations. During training, we set this tolerance $\varepsilon$ of forward and backward passes to $\varepsilon = \sqrt{T} \cdot 10^{-5}$ and $\sqrt{T} \cdot 10^{-8}$, respectively. At inference, we relax the tolerance to $\varepsilon = \sqrt{T} \cdot 10^{-2}$ (or we can use a smaller maximum iteration limit for Broyden's method; see discussions later). For the DEQ-TrellisNet instantiation, we roughly follow the settings of [8]. For DEQ-Transformers, we employ the relative positional embedding [16], with sequences of length 150 at both training and inference on the WikiText-103 dataset. Implementations and pretrained models can be found at `https://github.com/locuslab/deq`.

## 5.1 Copy Memory Task

The goal of the *copy memory task* is simple: to explicitly test a sequence model's ability to exactly memorize elements across a long period of time (see Appendix F). As shown in Table 1, DEQ demonstrates good memory retention over relatively long sequences ($T = 400$), with substantially better results than recurrent architectures such as LSTM/GRU (consistent with the findings in [7]).

## 5.2 Large-Scale Language Modeling

One issue encountered in prior works that take a continuous view of deep networks [11, 24] is the challenge of scaling these approaches to real, high-dimensional, large-scale datasets. In this subsection, we evaluate the DEQ approach on some large-scale language datasets and investigate its effectiveness as a practical "implicit-depth" sequence model.

**Performance on Penn Treebank.** Following the set of hyperparameters used by [8] for TrellisNet, we evaluate the DEQ-TrellisNet instantiation on word-level language modeling with the PTB corpus. Note that without an explicit notion of "layer", we do not add auxiliary losses, as was done in [8]. As shown in Table 2, when trained from scratch, the DEQ-TrellisNet achieves a test perplexity on par with the original deeply supervised TrellisNet.

**Performance on WikiText-103.** On the much larger scale WT103 corpus (about 100x larger than PTB), the DEQ-TrellisNet achieves better test perplexity than the original deep TrellisNet. For the Transformer instantiation, we follow the design of the Transformer-XL model [16]. We specifically compare to a "medium" Transformer-XL model (the largest released model that can fit on GPUs)

Table 3: DEQ-based models are competitive with SOTA deep networks of the same model size on the WikiText-103 corpus, with significantly less memory. †See Table 2 for more details on the memory benchmarking. Transformer-XL models are not weight-tied, unless specified otherwise.

| Word-level Language Modeling w/ WikiText-103 (WT103) | | | | |
|---|---|---|---|---|
| Model | # Params | Non-Embedding Model Size | Test perplexity | Memory† |
| Generic TCN [7] | 150M | 34M | 45.2 | - |
| Gated Linear ConvNet [17] | 230M | - | 37.2 | - |
| AWD-QRNN [33] | 159M | 51M | 33.0 | 7.1GB |
| Relational Memory Core [40] | 195M | 60M | 31.6 | - |
| Transformer-XL (X-large, adaptive embed., on TPU) [16] | 257M | 224M | **18.7** | 12.0GB |
| 70-layer TrellisNet (+ auxiliary loss, etc.) [8] | 180M | 45M | 29.2 | 24.7GB |
| 70-layer TrellisNet with *gradient checkpointing* | 180M | 45M | 29.2 | 5.2GB |
| **DEQ-TrellisNet (ours)** | 180M | 45M | **29.0** | **3.3GB** |
| Transformer-XL (medium, 16 layers) | 165M | 44M | 24.3 | 8.5GB |
| **DEQ-Transformer (medium, ours)**. | 172M | 43M | 24.2 | **2.7GB** |
| Transformer-XL (medium, 18 layers, adaptive embed.) | 110M | 72M | 23.6 | 9.0GB |
| **DEQ-Transformer (medium, adaptive embed., ours)** | 110M | 70M | **23.2** | 3.7GB |
| Transformer-XL (small, 4 layers) | 139M | 4.9M | 35.8 | 4.8GB |
| Transformer-XL (small, weight-tied 16 layers) | 138M | 4.5M | 34.9 | 6.8GB |
| **DEQ-Transformer (small, ours)**. | 138M | 4.5M | **32.4** | **1.1GB** |

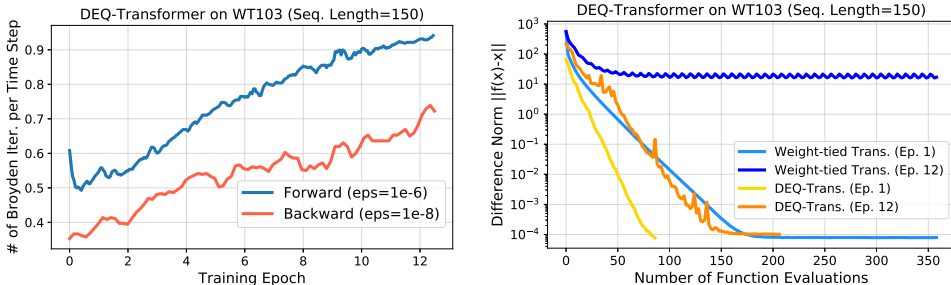

Figure 2: Left: number of Broyden iterations in forward and backward passes gradually grows with epochs. Right: DEQ-Transformer finds the equilibrium in a stable and efficient manner (whereas the deep transformer could oscillate around the fixed point, even when one exists).

and a "small" Transformer-XL model, while noting that the largest Transformer-XL network has massive memory requirements (due in part to very wide hidden features, batch sizes, and training-time sequence lengths, which would not be decreased by a DEQ) and can only be trained on TPUs [16]. In Table 3, we show that the DEQs yield competitive performance, outperforming prior SOTA approaches such as [16] on similar model sizes while consuming much less memory during training.

**Memory footprint of DEQ.** For conventional deep networks with $L$ layers, the training memory complexity is $O(L)$ since all intermediate activations are stored for backpropagation. In comparison, DEQs have an $O(1)$ (i.e., constant) memory footprint due to the root-finding formulation. We benchmark the reduced memory consumption in the last column of Tables 2 and 3, with controlled sequence lengths and batch sizes for fairness. On both instantiations, the DEQ approach leads to an over 80% (up to 88%) reduction in memory consumption by the model (excluding word embeddings, which are orthogonal to the comparison here). Moreover, we empirically verify (using a 70-layer TrellisNet) that DEQ consumes even less memory than gradient checkpointing [12], a popular technique that reduces the memory required to train a layer-based model to $O(\sqrt{L})$. Note that the DEQ's memory footprint remains competitive even when compared with baselines that are not weight-tied (a reduction of over 60%), with similar or better accuracy.

**Initialization of DEQ.** To train DEQ models, it is critical to ensure that the model is stable, such that the equilibrium state can be reliably approximated via quasi-Newton methods. While we found that the most commonly used initialization schemes with small values (around 0) suffice, it is generally important to make sure that DEQ starts with a small operator norm in the weight matrices. For both DEQ-TrellisNet and DEQ-Transformer, we observe that they are not sensitive to any specific initialization scheme since non-linearities such as $\sigma$/tanh and LayerNorm also help make $f_\theta$ contractive (and stable). We initialize the parameters of $f_\theta$ by sampling from $\mathcal{N}(0, 0.05)$.

Table 4: Runtime ratios between DEQs and corresponding deep networks at training and inference ($> 1\times$ implies DEQ is slower). The ratios are benchmarked on WikiText-103.

| | DEQ / 18-layer Transformer | | DEQ / 70-layer TrellisNet | |
| --- | --- | --- | --- | --- |
| | Training | Inference | Training | Inference |
| | $2.82\times$ | $1.76\times$ | $2.40\times$ | $1.64\times$ |

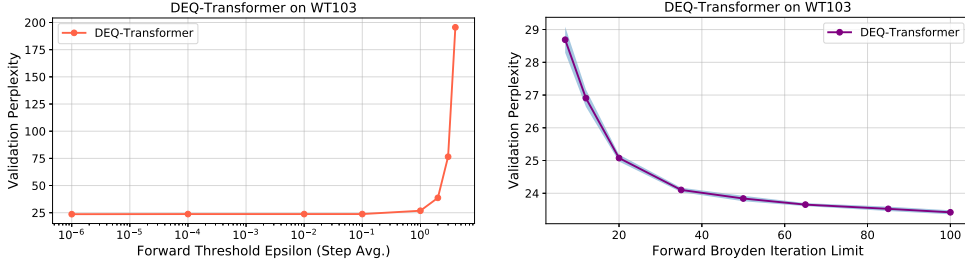

Figure 3: DEQ can be accelerated by leveraging higher tolerance $\varepsilon$ (left) or a lower Broyden iteration limit (right). In general, poor estimates of the equilibrium can hurt DEQ performances.

**Convergence to equilibrium.** The deep equilibrium model does not have "layers". One factor that affects computation time in DEQs is the number of Broyden iterations in forward/backward passes, where each forward Broyden step evaluates $f_\theta$ once, and a backward step computes a vector-Jacobian product. We find that in general the number of Broyden iterations gradually increases with training epochs (Figure 2, left, where the $y$-axis is computed by $\frac{\text{Total Broyden Iterations}}{\text{Sequence Length}}$), an observation similar to the one reported for training Neural ODEs [11]. One factor contributing to this phenomenon could be that the training pushes the operator norm of $J_{f_\theta}$ to larger values, making the fixed point harder to solve. Meanwhile, the backward pass requires much fewer iterations than the forward, primarily due to the simplicity of the linear system in Eq. (11). We also find that DEQs can almost always converge to the sequence-level fixed point, much more efficiently than original weight-tied transformers (Figure 2, right). Note that after 12 epochs, deeply stacked self-attention tends to oscillate around the fixed point, while DEQs exhibit stable convergence with the quasi-Newton method.

**Broyden iterations and the runtime of DEQ.** Unlike conventional deep networks that come with a fixed number $L$ of layers, the runtime of DEQ depends strongly on the number of Broyden steps to reach the equilibrium. Therefore, it's challenging to fairly compare the runtimes of implicit-depth models like DEQ with those of corresponding weight-tied deep networks (e.g., using higher depth necessarily takes longer to run). Ideally, the values of $\varepsilon$ should be as small as possible so as to ensure that the analytical gradients from Theorem 1 are accurate. However, we empirically observe that using a higher $\varepsilon$ or a lower iteration limit allows the DEQ to be trained and evaluated much faster with only a small degradation in performance. For instance, generally we find $\varepsilon < 0.1$ or an iteration limit of 30 (on sequence length 75) to be sufficient for competitive performance. Figure 3 visualizes this tradeoff on a medium DEQ-Transformer (without adaptive embedding). Note that accuracy quickly diverges when tolerance $\varepsilon$ is too large (Figure 3, left), suggesting that a poor estimate of the equilibrium can hurt DEQ performances. Table 4 provides approximate runtimes for competitive-accuracy DEQs on WikiText-103. DEQs are typically slower than layer-based deep networks.

Additional empirical remarks as well as training tips are provided in Appendix E.

## 6 Conclusion

Deep networks have predominantly taken the form of stacks of layers. We propose the deep equilibrium approach (DEQ), which models temporal data by directly solving for the sequence-level fixed point and optimizing this equilibrium for better representations. DEQ needs only $O(1)$ memory at training time, is agnostic to the choice of the root solver in the forward pass, and is sufficiently versatile to subsume drastically different architectural choices. Our experiments have shown that DEQs have good temporal memory retention, are able to scale to realistic, large-scale sequence tasks, and perform competitively with, or slightly outperform, SOTA methods. Overall, we believe that the DEQ approach provides an interesting and practical new perspective on designing and optimizing sequence models.

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
