[Supplementary Material · deq_supp.pdf]

# A Backward Pass of the Deep Equilibrium Model

One of the core benefits of the DEQ approach comes from its analytical backward gradient at equilibrium. In this section, we provide a proof to Theorem 1 (which we restate here).

**Theorem 1.** *(Gradient of the Equilibrium Model)* *Let $\mathbf{z}_{1:T}^\star \in \mathbb{R}^{T \times d}$ be an equilibrium hidden sequence with length $T$ and dimensionality $d$, and $\mathbf{y}_{1:T} \in \mathbb{R}^{T \times q}$ the ground-truth (target) sequence. Let $h : \mathbb{R}^d \to \mathbb{R}^q$ be any differentiable function and $\mathcal{L} : \mathbb{R}^q \times \mathbb{R}^q \to \mathbb{R}$ be a loss function (where $h, \mathcal{L}$ are applied in vectorized manner) that computes*

$$\ell = \mathcal{L}(h(\mathbf{z}_{1:T}^\star), \mathbf{y}_{1:T}) = \mathcal{L}(h(\mathsf{RootFind}(g_\theta; \mathbf{x}_{1:T})), \mathbf{y}_{1:T}). \tag{13}$$

*Then the loss gradient w.r.t.* $(\cdot)$ *(for instance, $\theta$ or $\mathbf{x}_{1:T}$) is*

$$\frac{\partial \ell}{\partial (\cdot)} = -\frac{\partial \ell}{\partial \mathbf{z}_{1:T}^\star} \left( J_{g_\theta}^{-1} \big|_{\mathbf{z}_{1:T}^\star} \right) \frac{\partial f_\theta(\mathbf{z}_{1:T}^\star; \mathbf{x}_{1:T})}{\partial (\cdot)} = -\frac{\partial \ell}{\partial h} \frac{\partial h}{\partial \mathbf{z}_{1:T}^\star} \left( J_{g_\theta}^{-1} \big|_{\mathbf{z}_{1:T}^\star} \right) \frac{\partial f_\theta(\mathbf{z}_{1:T}^\star; \mathbf{x}_{1:T})}{\partial (\cdot)}, \tag{14}$$

*where $J_{g_\theta}^{-1} \big|_{\mathbf{x}}$ is the inverse Jacobian of $g_\theta$ evaluated at $\mathbf{x}$.*

*Proof of Theorem 1.* We first write out the equilibrium sequence condition: $f_\theta(\mathbf{z}_{1:T}^\star; \mathbf{x}_{1:T}) = \mathbf{z}_{1:T}^\star$. By implicitly differentiating two sides of this condition with respect to $(\cdot)$:

$$\frac{\mathrm{d}\mathbf{z}_{1:T}^\star}{\mathrm{d}(\cdot)} = \frac{\mathrm{d}f_\theta(\mathbf{z}_{1:T}^\star; \mathbf{x}_{1:T})}{\mathrm{d}(\cdot)} = \frac{\partial f_\theta(\mathbf{z}_{1:T}^\star; \mathbf{x}_{1:T})}{\partial (\cdot)} + \frac{\partial f_\theta(\mathbf{z}_{1:T}^\star; \mathbf{x}_{1:T})}{\partial \mathbf{z}_{1:T}^\star} \frac{\mathrm{d}\mathbf{z}_{1:T}^\star}{\mathrm{d}(\cdot)}$$

$$\implies \left( I - \frac{\partial f_\theta(\mathbf{z}_{1:T}^\star; \mathbf{x}_{1:T})}{\partial \mathbf{z}_{1:T}^\star} \right) \frac{\mathrm{d}\mathbf{z}_{1:T}^\star}{\mathrm{d}(\cdot)} = \frac{\partial f_\theta(\mathbf{z}_{1:T}^\star; \mathbf{x}_{1:T})}{\partial (\cdot)}$$

Since $g_\theta(\mathbf{z}_{1:T}^\star) = f_\theta(\mathbf{z}_{1:T}^\star; \mathbf{x}_{1:T}) - \mathbf{z}_{1:T}^\star$, we have

$$J_{g_\theta} \big|_{\mathbf{z}_{1:T}^\star} = -\left( I - \frac{\partial f_\theta(\mathbf{z}_{1:T}^\star; \mathbf{x}_{1:T})}{\partial \mathbf{z}_{1:T}^\star} \right),$$

which implies

$$\frac{\partial \ell}{\partial (\cdot)} = \frac{\partial \ell}{\partial \mathbf{z}_{1:T}^\star} \frac{\mathrm{d}\mathbf{z}_{1:T}^\star}{\mathrm{d}(\cdot)} = -\frac{\partial \ell}{\partial \mathbf{z}_{1:T}^\star} \left( J_{g_\theta}^{-1} \big|_{\mathbf{z}_{1:T}^\star} \right) \frac{\partial f_\theta(\mathbf{z}_{1:T}^\star; \mathbf{x}_{1:T})}{\partial (\cdot)}.$$

$\square$

# B Sufficiency of a Single DEQ "Layer"

A hypothetical extension to the DEQ idea follows from the "deep" philosophy: if one DEQ works so well, why don't we stack multiple DEQ modules with different parameters $f_{\theta^{[i]}}$ $(i = 1, 2, \dots)$? We (re-)state and prove the following theorem, which demonstrates the universality of the DEQ model (i.e., sufficiency of exactly one DEQ "layer").

**Theorem 2.** *(Universality of "Single-layer" DEQs)* *Let $\mathbf{x}_{1:T} \in \mathbb{R}^{T \times p}$ be the input sequence, and $\theta^{[1]}, \theta^{[2]}$ the sets of parameters for stable transformations $f_{\theta^{[1]}} : \mathbb{R}^r \times \mathbb{R}^p \to \mathbb{R}^r$ and $v_{\theta^{[2]}} : \mathbb{R}^d \times \mathbb{R}^r \to \mathbb{R}^d$, respectively. Then there exists $\Gamma_\Theta : \mathbb{R}^{d+r} \times \mathbb{R}^p \to \mathbb{R}^{d+r}$, where $\Theta = \theta^{[1]} \cup \theta^{[2]}$ s.t.*

$$\mathbf{z}_{1:T}^\star = \mathsf{RootFind}\big(g_{\theta^{[2]}}^v; \mathsf{RootFind}\big(g_{\theta^{[1]}}^f; \mathbf{x}_{1:T}\big)\big) = \mathsf{RootFind}\big(g_\Theta^\Gamma; \mathbf{x}_{1:T}\big)_{[:, -d:]} \tag{15}$$

*where $[\cdot]_{[:, -d:]}$ denotes the last $d$ feature dimensions of $[\cdot]$.*

*Proof of Theorem 2.* Assume $\mathbf{z}_{1:T}^{[1]\star} = \mathsf{RootFind}\big(g_{\theta^{[1]}}^f; \mathbf{x}_{1:T}\big) \in \mathbb{R}^r$ is the equilibrium of the first DEQ module under transformation $f_{\theta^{[1]}}$. Define $\Theta = \theta^{[1]} \cup \theta^{[2]}$, and $\Gamma_\Theta(\mathbf{w}_{1:T}; \mathbf{x}_{1:T}) : \mathbb{R}^{d+r} \times \mathbb{R}^p \to \mathbb{R}^{d+r}$ by:

$$\Gamma_\Theta(\mathbf{w}_{1:T}; \mathbf{x}_{1:T}) = \Gamma_\Theta\left( \begin{bmatrix} \mathbf{w}_{1:T}^{(1)} \\ \mathbf{w}_{1:T}^{(2)} \end{bmatrix}; \mathbf{x}_{1:T} \right) = \begin{bmatrix} f_{\theta^{[1]}}(\mathbf{w}_{1:T}^{(1)}, \mathbf{x}_{1:T}) \\ v_{\theta^{[2]}}(\mathbf{w}_{1:T}^{(2)}, \mathbf{w}_{1:T}^{(1)}) \end{bmatrix} \tag{16}$$

Then $\mathbf{w}_{1:T}^\star = \begin{bmatrix} \mathbf{z}_{1:T}^{[1]\star} \\ \mathbf{z}_{1:T}^\star \end{bmatrix}$ is a fixed point of $\Gamma_\Theta(\cdot; \mathbf{x}_{1:T})$, which completes the proof. $\square$

## C   Universality of Weight-tied, Input-injected Networks

Although the DEQ model corresponds to an infinite-depth network, as mentioned above it applies only to the specific case of *weight-tied*, input-injected infinite-depth models. This seems at first glance a substantial restriction over traditional deep networks, which have no requirement that the weights at each layer be identical. However, as we show below, this is not an actual restriction on the representational capacity from a mathematical point of view. Specifically, any deep network can be represented as a deep weight-tied network with no increase in depth and only a linear increase in the size of the hidden layer. This argument is equivalent to that presented in the TrellisNet work [8, Theorem 1], but we include it here in a slightly simpler and more general form. We emphasize that in practice we do *not* use the sparse structure below to construct the weight-tied layers for DEQ, but instead just use dense matrices $W_z$ and $W_x$. However, the theorem below is important in establishing that there is no notable representational loss.

**Theorem 3.** *(**Universality of Weight-tied Deep Networks**) Consider a traditional L-layer deep network defined by the relation*

$$\mathbf{z}^{[i+1]} = \sigma^{[i]}(W^{[i]}\mathbf{z}^{[i]} + \mathbf{b}^{[i]}), \quad i = 0, \dots, L-1, \quad \mathbf{z}^{[0]} = \mathbf{x} \tag{17}$$

*where $\mathbf{z}^{[i]}$ denotes the hidden features at depth $i$, $W^{[i]}$, $\mathbf{b}^{[i]}$ are parameters of the network, $\sigma^{[i]}$ is the non-linearity at depth $i$, and $\mathbf{x}$ is the original input. Then the same network can be represented by a weight-tied, input-injected network of equivalent depth*

$$\tilde{\mathbf{z}}^{[i+1]} = \sigma(W_z\tilde{\mathbf{z}}^{[i]} + W_x\mathbf{x} + \tilde{\mathbf{b}}), \quad i = 0, \dots, L-1. \tag{18}$$

*where $\sigma$, $W_z$, $W_x$ and $\tilde{\mathbf{b}}$ are constant over all layers.*

*Proof of Theorem 3.*  The proof is constructive: we build the weight-tied network equivalent to the original network by contructing the relevant matrices using a simple "shift" operation. In particular, we define the network parameters as

$$W_z = \begin{bmatrix} 0 & 0 & \dots & 0 & 0 \\ W^{[1]} & 0 & \dots & 0 & 0 \\ 0 & W^{[2]} & \dots & 0 & 0 \\ \vdots & \vdots & \ddots & \vdots & \vdots \\ 0 & 0 & \dots & W^{[L-1]} & 0 \end{bmatrix}, W_x = \begin{bmatrix} W^{[0]} \\ 0 \\ \vdots \\ 0 \end{bmatrix}, \tilde{\mathbf{b}} = \begin{bmatrix} \mathbf{b}^{[0]} \\ \mathbf{b}^{[1]} \\ \vdots \\ \mathbf{b}^{[L-1]} \end{bmatrix}, \sigma = \begin{bmatrix} \sigma^{[0]} \\ \sigma^{[1]} \\ \vdots \\ \sigma^{[L-1]} \end{bmatrix}. \tag{19}$$

It is clear from inspection that after $L$ applications of the layer, i.e.,

$$\tilde{\mathbf{z}}^{[i+1]} = \sigma(W_z\tilde{\mathbf{z}}^{[i]} + W_x\mathbf{x} + \tilde{\mathbf{b}}) \tag{20}$$

using these parameters the hidden vector $\tilde{\mathbf{z}}$ will take on the value

$$\tilde{\mathbf{z}}^{[L]} = \begin{bmatrix} \mathbf{z}^{[1]} \\ \mathbf{z}^{[2]} \\ \vdots \\ \mathbf{z}^{[L]} \end{bmatrix}. \tag{21}$$

Thus the weight-tied network computes all the same terms as the original network, using the same depth as the original network, and with a hidden unit size that is just the sum of the individual hidden unit sizes in the original network. This establishes the claim of the theorem.  □

## D   Empirical Convergence of Weight-tied Deep Nets

As mentioned in Section 3, one motivation for optimizing the sequence-level equilibrium comes from our empirical observations that, starting at some point of the deep stacking, weight-tied deep networks *begin* to converge to a fixed point. We show in Figure 4 the convergence of trained layer-based TrellisNet (weight-tied temporal convolutions) and universal transformer (weight-tied self-attention) on randomly selected test samples of different lengths $T =$ 100, 200, 400 and 800. In both cases, we see a tendency of the activations to converge. Notably, for transformers, we find stacked iterations

Figure 4: The convergence of intermediate activations in TrellisNet (with kernel size 2) and weight-tied transformers on different sequence lengths.

usually lead to a oscillatory behavior on the plots (Figure 4), with values fluctuating around the actual fixed point (which we empirically verify can be found much more easily with Newton or quasi-Newton methods).

In practice, due to limited computation, we usually set the number of layers to a predetermined number (e.g., 60 layers) and rarely reach the inference depths analyzed in Figure 4. Moreover, in the specific case of transformers, [1] stablizes the training of deep transformers (64-layer) on character-level language modeling with the help of various strong auxiliary losses at intermediate layers. In a certain sense, the addition of auxiliary losses have a similar effect as solving for equilibrium: we want intermediate-level hidden units to be both close to the target and as stable as possible (without drastic interlayer differences).

## E  More Remarks on DEQ

**Applicability of other deep techniques.**   While the DEQ approach does not preclude specific architectural choices of $f_\theta$ (which means techniques such as layer normalization [5] or weight normalization [39] can work as is), it is not clear how certain regularizations such as auxiliary losses [46, 8] could be applied on DEQ, since there are no more "layers". For dropout [44], we follow the practice of [8], which adapts the RNN variational dropout [22] scheme to feedforward networks by applying the same mask at all levels. We empirically find that adding dropout makes the quasi-Newton methods slower to converge (i.e., inference-time equilibria are easier to find without the presence of noisy zeros). Since the weights of $f_\theta$ (and thus its operator norm) are directly related to the stability of root-finding, we observe that weight normalization [39] typically finds more stable parameters and slows down the growth of forward/backward Broyden's iterations (as described in Figure 2).

**Imbalances within minibatches.**   Not all sequences in a minibatch converge to the equilibrium with the same number of iterations. However, with standard batched CUDA operations, the sequences that converge faster essentially need to "wait" for the slower ones. Though we empirically find such imbalance to be relatively small in scale, it could mean an inefficient GPU utilization at times.

**Warmup of DEQ models with shallow nets.**   Instead of training the DEQ from scratch, empirically we find that one can accelerate the DEQ training by pretraining a shallow weight-tied stack of $f_\theta$ (e.g., 2 layers), and using the resulting parameters to initialize the DEQ. In general, a shallow model plateaus at much lower accuracy than corresponding DEQs or deeper weight-tied networks. However, given the very small number of layers, a shallow model offers a memory- and computation-efficient starting point for DEQ training.

**Training DEQs with subsequences.**   On extremely long sequences (e.g., $T > 1000$), the forward-pass fixed points can be challenging to solve accurately (especially at the start of the training) even with the help of the root-finding methods. Therefore, in practice, we suggest breaking these long sequences into a few subsequences when needed (recall that the forward pass can be *any* black-box root-finder). Moreover, with the help of Theorem 1, such subsequence technique can be used in the backward pass as well (where we solve for Eq. (11)). For instance, on a sequence

$\mathbf{z}^\star_{1:T} = \begin{bmatrix} \mathbf{z}^\star_{1:(T/2)} & \mathbf{z}^\star_{(T/2):T} \end{bmatrix}$:

$$\frac{\partial \ell}{\partial (\cdot)} = \frac{\partial \ell}{\partial \mathbf{z}^\star_{(T/2):T}} \underbrace{\frac{\partial \mathbf{z}^\star_{(T/2):T}}{\partial (\cdot)}}_{(A)} + \frac{\partial \ell}{\partial \mathbf{z}^\star_{(T/2):T}} \underbrace{\frac{\partial \mathbf{z}^\star_{(T/2):T}}{\partial \mathbf{z}^\star_{1:(T/2)}}}_{(B)} \underbrace{\frac{d\mathbf{z}^\star_{1:(T/2)}}{d(\cdot)}}_{(C)} \tag{22}$$

where terms (A) and (B) require one evaluation of $f_\theta(\mathbf{z}^\star_{(T/2):T}; [\mathbf{x}_{(T/2):T}, \mathbf{z}^\star_{1:(T/2)}])$ and term (C) requires one evaluation of $f_\theta(\mathbf{z}^\star_{1:(T/2)}; \mathbf{x}_{1:(T/2)})$. Hence, the memory cost is equivalent to that of applying $f_\theta$ once on the entire $\mathbf{z}^\star_{1:T}$ (but with the subsequences' equilibrium likely easier to optimize).

## F  Task Descriptions

We briefly introduce the three sequence prediction tasks/datasets that we employ to evaluate the DEQ approach in Section 5.

**Copy memory task.**  The copy memory task is a small but challenging synthetic stress test that has been frequently used in prior work to test a sequence model's memory retention ability [53, 4, 7]. In this task, each sequence $\mathbf{x}_{1:(T+20)}$ is 1-dimensional and has length $T + 20$, with $\mathbf{x}_{1:10}$ randomly selected from integers $1, 2, \ldots, 8$ (with repetition). The rest of the input elements are all filled with zeros, except for the $\mathbf{x}_{T+10} = 9$. The goal of this task is to produce $\mathbf{y}_{1:(T+20)}$ such that $\mathbf{y}_{1:T+10} = \mathbf{0}$ and $\mathbf{y}_{T+11:T+20} = \mathbf{x}_{1:10}$. In other words, a sequence model trained on this task is expected to "recall" the first 10 elements of the sequence once it sees the delimiter $\mathbf{x}_{T+10} = 9$, and copy them to the end of the sequence. We generate 20K training samples and 2K testing samples. In prior works, [7] have shown that RNNs generally struggle with the task, especially when $T > 100$, whereas feedforward models tend to have better memory.

**Penn Treebank.**  The Penn Treebank (PTB) corpus [31] is a commonly used dataset for character- and word-level language modeling. When used for word-level language modeling, PTB contains about 888K words at training, with a vocabulary size of 10,000. As this is a comparatively small language corpus (with punctuations and capitalization removed), prior work has shown that well-designed regularizations are required for best results [34, 52].

**WikiText-103.**  The training corpus of WikiText-103 (WT103) [35] is about 110 times larger than PTB, with a vocabulary size over 260K. In general, this dataset is considered much more realistic than many others because it contains many rare words and retains punctuation, numbers, and capitalization from the original Wikipedia articles. WT103 is thus used to evaluate how well a sequence model scales to long sequences from a large vocabulary. This dataset has been frequently used in recent work with high-capacity sequence models [9, 8, 16, 6].