[Reviews · NeurIPS 2019]

Reviewer 1



==== Based on the authors response, I find the comparison against gradient checkpointing they provide satisfactory. Please ensure it is included in the final draft ==== This work considers handling sequences of networks layers with identical weights (i.e. weight tied layers) via fixed point computation. Instead of directly computing the sequence, a quasi-newton method is used to approximate the fixed point of the sequence. This has the advantage that the gradient has a simpler form, although one which must also be computed iteratively. The advantages are: • Much lower memory usage as intermediate tensors do not need to be stored for use in the backwards pass. Approximately 4-10x lower for the considered models. • Empirically (sometimes) better perplexity compared to iterating. The disadvantages are: • About 3-5 times slower to train, 1.7-2x slower at evaluation time. • Significant additional implementation complexity • If there is no fixed point the method may not work. The paper is well-written and generally polished. I'm disappointed that the runtime comparisons are relegated to the appendix, it is poor scholarship to hide the disadvantages of your method. Reducing the memory requirements of language models is an important goal, as they are typically memory constrained during training for the largest SOTA models. A practical approach here could have significant impact. The experiments seem reasonable, the authors compare against current SOTA level models. It's unclear if there is any advantage compared to using gradient checkpointing. Checkpointing is discussed in the related work section but not tested against. Checkpointing requires only 1 extra forward pass, which is typically a 1.5x overhead, although the memory reduction may not be as significant (depending on the exact network architecture). The additional computational resources saved could be used to train a slightly larger model, which may reach similar perplexities. If this paper clearly showed that the approach was superior to checkpointing, it would be a much stronger submission.

Reviewer 2



I have read the rebuttal and the other reviews and maintain my score. I look forward to discussion of the (apparently since improved) training time in the main text. ======== The submission concisely presents a simple but powerful idea that will have impact. The DEQ model is a sensible model that draws on neural ODE-type ideas but stands on its own. (To my knowledge this is the first application of implicit depth ideas to sequence modeling). The derived algorithm is sensible, clearly explained, and the proofs appear correct (I checked them). The experiments seem reasonable and rigorous. The paper is well-written and exceptionally easy to follow. Overall I strongly recommend accepting this paper. A few questions: - in the paper, the authors state (and empirically show in the Appendix) that they can get away with a large \eps tolerance at inference time. How did they decide upon the \eps=1e-6, 1e-8 choices for training? (which one would expect may change the quality of the learned model) - How were the models initialized? I would expect this has substantial influence on the stability of training, since it may affect e.g. whether f is non-expansive. (From the code it looks like the initialization is pretty standard but this should be discussed in the main text) - the current version of the paper defers all discussion of runtime to the Appendix. Since the (slightly) slower runtime is a drawback of the model, it would be more fair to state this in the main text.

Reviewer 3



I have read the author response and am satisfied with it. As the other reviewers have pointed out, it would be good to move the discussion on training time to the main text. ----------------------------------------------------------------------------------------------------------------------------- This paper proposes a new approach to modeling sequential data called Deep Equilibrium Models (DEQ) whereby instead of using a weight tied deep neural network its fixed point is directly computed. Results show that this approach still gives the same perplexity as the deep neural network but has much lesser memory footprint. However the inference and training cost could be 2-4x times higher. The paper is written and organized nicely and is easy to follow. The idea itself seems novel and addresses an important problem of reducing the memory footprint of deep neural networks. However I am not sure what happens when the fixed point is unstable? Nevertheless the authors have done extensive experimentation both on synthetic and real-world datasets and have shown that DEQ reduces the memory footprint by more than 80% while giving the same perplexity.

[Author Response · NeurIPS 2019]

**General Comments.** We thank the reviewers for their valuable feedback. As the reviewers point out, the deep equilibrium model offers a new perspective on deep networks. Instead of actually creating a deep stack of layers, the central idea of this paper is to develop an alternative view of deep learning where we directly optimize for and backpropagate through an equilibrium state of the network (which, to the best of our knowledge, no deep approaches have explored or targeted to date, and similar ideas such as the Neural ODE differ significantly in their formulation). The way DEQ "ignores" depth and solves for the equilibrium suggests a different view of output modeling and further leads to certain interesting properties beyond the obvious reductions in memory footprint (cf. Theorem 1 and 2).

Importantly, compared to prior implicit-depth approaches such as Neural ODEs, in this work we also demonstrate the potential power and applicability of such models on practical, large-scale and high-dimensional datasets. In fact, we are able to get 24.0 ppl using a slightly larger DEQ-Transformer than in Table 3, which outperforms the current SOTA result that can run on GPUs (these results will be reflected in the revision). We believe that the equilibrium view of deep learning could lead to many directions of research, in both designing better sequence models (e.g., via better-designed $f_\theta$, see Theorem 2) and studying the properties of the equilibrium optimization.

We also agree with the reviewers that the runtime discussion should be moved into the main text. We briefly include some of our observations below and will have a more thorough analysis of the relationship between threshold $\varepsilon$, training/inference speeds, and modeling accuracy in the experiment section of the revision. We now address specific questions/comments raised by each reviewer.

**Review #1.** We thank reviewer #1 for the valuable feedback. As we highlighted in the general comments above, the DEQ approach is very different from techniques like gradient checkpointing (GC). In essence, GC enables training an $L$-layer network using $O(\sqrt{L})$ memory, without actually affecting the computations themselves (GC only recomputes certain blocks). It is an implementation-based methodology that is practical on almost any layer-based network. On the other hand, continuous/implicit-depth models such as Neural ODE and DEQ reduce memory requirements by *formulation* rather than implementation, as these models usually come from certain black-box solvers and analytical backpropagation.

Quantitatively, we have followed the reviewer's suggestion and compared GC and DEQ using a 70-layer TrellisNet (w/ aux. loss, etc.) on WT103. We find that GC works best when we checkpoint after every 9 layers, and record a 5.2GB memory footprint at training time under these conditions. This is 57% more than the DEQ memory footprint (see Table 3). The training speed of GC is approximately $1.6\times$ slower than original training, while DEQ can be up to $2.4\times$ slower (this is an updated result, see our response to reviewer #3 for more details). More fundamentally, though, we should emphasize GC offers $O(\sqrt{L})$ memory consumption while DEQ is $O(1)$. (And recall we are dealing with $L \to \infty$ ...)

**Review #3.** We thank reviewer #3 for the comments, and for taking the time to check our proof and read our code. We also feel that DEQ provides a new and exciting direction for further designing better implicit-depth models as well as exploring the properties of equilibrium training.

We originally picked the values of $\varepsilon$ just to ensure that we get a fixed point that is as accurate as possible under the superlinear convergence of quasi-Newton methods. However, since the submission we have further observed that the conclusion from Figure 4 also holds in training. By using larger $\varepsilon$ or a smaller iteration limit, we find that the model can be trained much faster with only a small degradation in performance (e.g., we get 24.3 ppl on WT103 with DEQ-Transformer by limiting the max # of Broyden iterations to 35; the same model can yield 24.0 ppl using a smaller $\varepsilon$). We generally find that $\varepsilon < 0.01$ is sufficient. With that observation, the DEQ training time is now around 2-2.4$\times$ that of the original networks (see Table 4) without materially affecting accuracy. We are running more settings and will provide a detailed discussion of this (and some caveats) in the revision.

Regarding initialization, we find that most commonly used initialization schemes with small values (around 0) should suffice. It is important to ensure that the model starts with a small operator norm in the weight matrices. DEQ is not sensitive to any specific initialization scheme because non-linearities such as $\sigma$/tanh and LayerNorm help make $f_\theta$ contractive (and stable). We are happy to discuss this in the revision.

**Review #4.** We thank reviewer #4 for the comments. DEQ essentially provides a way to model a deep network at its infinite limit. Deep learning research indicates that more layers frequently lead to better results, and DEQ provides a way to explore the limits of layer stacking without paying an exorbitant price in memory or computation. As highlighted in our response to reviewer #3, we can also further reduce the training/inference cost by less accurate (but still good enough) fixed-point estimation using larger $\varepsilon$, which can be an interesting topic for further research. In addition, while we did not observe any "unstable" fixed points empirically, we believe it is important to ensure that the transformation $f_\theta$ itself is stable and contractive (e.g., ideally, having $J_{f_\theta}$ operator norm less than 1 would be a sufficient condition).

[Meta-Review · NeurIPS 2019]

This paper proposes a novel technique to reduce memory consumption for training deep sequence models by using fixed point formulation. The technique requires constant memory regardless of the effective depth of the network. All the reviewers have found the work to be of sufficient novelty and interest to have it published at NeurIPS. The algorithms are theoretically justified. Overall the paper is well-written and easy to follow. The authors’ rebuttal has addressed the reviewers’ concern around comparison with gradient checkpointing, However, the proposed method is much slower in training as well as inference which may restrict the practical utility of the work for very large datasets. Pls include this discussion in the revised version.